# Network topology and movement cost, not updating mechanism, determine the evolution of cooperation in mobile structured populations

**Diogo L. Pires[1], Igor V. Erovenko[2], Mark Broom[1] ***

**1** Department of Mathematics, University of London, London, United Kingdom, **2** Department of Mathematics and Statistics, University of North Carolina at Greensboro, Greensboro, NC, United States of America

\* Mark.Broom@city.ac.uk

**Data Availability Statement:** Data are available at https://figshare.com/projects/Network_topology_and_movement_cost_not_updating_mechanism_determine_the_evolution_of_cooperation_in_

## Abstract

Evolutionary models are used to study the self-organisation of collective action, often incorporating population structure due to its ubiquitous presence and long-known impact on emerging phenomena. We investigate the evolution of multiplayer cooperation in mobile structured populations, where individuals move strategically on networks and interact with those they meet in groups of variable size. We find that the evolution of multiplayer cooperation primarily depends on the network topology and movement cost while using different stochastic update rules seldom influences evolutionary outcomes. Cooperation robustly co-evolves with movement on complete networks and structure has a partially detrimental effect on it. These findings contrast an established principle from evolutionary graph theory that cooperation can only emerge under some update rules and if the average degree is lower than the reward-to-cost ratio and the network far from complete. We find that group-dependent movement erases the locality of interactions, suppresses the impact of evolutionary structural viscosity on the fitness of individuals, and leads to assortative behaviour that is much more powerful than viscosity in promoting cooperation. We analyse the differences remaining between update rules through a comparison of evolutionary outcomes and fixation probabilities.

## Introduction

The self-organisation of collective behaviour is observed in populations across all levels of complexity, from microorganisms [1–4] to human societies [5–7]. The modelling of evolutionary processes, often formulated within the framework of evolutionary game theory, has assisted the study of how these phenomena emerge, especially when conflicts may deem them counter-intuitive at first sight [8, 9]. These models take into consideration that the actions of individuals have mutually-impacting outcomes, thus leading to frequency-dependent fitness in evolving populations. Initial models often assumed infinite populations [8, 10, 11] due to the simplicity that this assumption offers and their resulting mathematical tractability.

mobile_structured_populations/171747 The DOI is: https://doi.org/10.6084/m9.figshare.23628423.

**Funding:** MB and DP have received funding from the European Union's Horizon 2020 research and innovation programme under the Marie Sklodowska-Curie grant agreement No 955708 https://rea.ec.europa.eu/funding-and-grants/horizon-europe-marie-sklodowska-curie-actions_en The funders had no role in study design, data collection and analysis, decision to publish, or preparation of the manuscript.

**Competing interests:** The authors have declared that no competing interests exist.

However, this choice leaves out the impact that the finiteness of real populations has on emerging behaviour [12, 13], and may make models inflexible to incorporating realistic aspects of populations [14]. Given this, a growing interest in finite population models emerged and established them as an essential part of the field [12, 13, 15].

Real populations are often observed to be structured in the sense that the interactions between individuals are not random and distinct connections may form [16–19]. At the same time, this feature has long been known to affect the outcome of evolutionary processes [20–22]. Subsequently, structure has been incorporated into evolutionary game-theoretic models by considering pairwise interaction networks of individuals in a population [23], termed evolutionary graph theory. In this context, population structure was found to be a promoter of cooperation when the reward-to-cost ratio exceeds the average number of neighbours each individual has and the network is far from complete [24]. This success was associated with the viscosity of evolutionary processes on graphs, which may only be seen under particular evolutionary dynamics such as the DBB and BDD dynamics (which will be defined precisely later). Nonetheless, the simplicity of the rule consolidated population structure as one fundamental mechanism for the evolution of cooperation [25], and motivated its study both mathematically [26–29] and in parallel with computational simulations [30–34].

Many collective action problems that individuals face require accounting for multiplayer interactions [35–40]. These can be approached considering interacting groups arising from realistic encounters of individuals moving on spatial or virtual networks, which can be formalised generally through the framework introduced in [41]. This framework leads to an emerging higher-order interaction network [42] which cannot be reduced to a graph when payoff functions have nonlinear dependencies on the number of individuals of each type in the interacting group, as happens often under public goods games [40].

In the context of this framework, several types of movement models have been studied, a review of which is provided in [43] together with an analysis of their robustness and applicability. Models with completely independent movement are defined as those under which individuals move independently of both their past positions and of each other's simultaneous movements [41, 44]. In this context, the territorial raider model was introduced in [41], and later simplified in [45] based on one single home fidelity parameter ($h$) for the whole population. In [46], six distinct evolutionary dynamics including the BDB [23], the DBB [24] and the LB [23] dynamics are expanded to allow their application to more general evolutionary models. Some of these dynamics were then explored in the context of the territorial raider model in small networks [45], networked subpopulations [46] and in complex networks with distinct structural properties [47, 48]. From extensive analysis of this whole set of results, it was concluded that only two of these six dynamics allowed structure to sustain the evolution of cooperation and only in some populations far from well-mixed [46], similarly to what occurs under pairwise interaction networks [24].

The landscape changes completely when we drop the assumption that movement is independent of previous positions. In [49], a Markov movement model is introduced, i.e. a model under which the next position of an individual depends only on positions at the current time step. It is shown that complete networks sustain the co-evolution of cooperation and assortative behaviour, allowing cooperators to find and stick to each other in groups until they are found by defectors. Even though defectors move around the network and find cooperating groups to exploit, they generally spend less time amongst them under positive movement costs. The chosen evolutionary dynamics were suggested to have a small impact on the results due to the symmetry and completeness of the studied evolutionary graph, an observation hypothesised to not translate into alternative topologies in [49]. Exploring circle and star networks, it is shown in [50] that non-complete topologies can be detrimental to the evolution of

cooperation under movement, potentially due to the negative impact of a lower clustering coefficient and a higher degree centralisation on the assortative behaviour described above.

The modelling framework considered has a number of factors in common with classical evolutionary graph theory models, but also introduces some differences. Evolutionary graph theory is composed of three core components: the underlying interaction structure (the graph), the evolutionary dynamics, and the game played between the participants. In the movement models mentioned above, the evolutionary dynamics are effectively the same set as the ones used in evolutionary graph theory, the only difference being that the weights used are calculated rather than just defined by a static interaction network. As our interactions can occur in groups of arbitrary size, we use multiplayer games, which are otherwise similar to those considered in other models. Evolutionary graph theory often uses multiplayer games, though interacting groups do not emerge naturally from their interactions as in our framework, but often involve all players neighbouring a particular individual as in [51].

The main difference between this framework and evolutionary graph theory is population structure. Each individual has a distribution over a number of places and different individual groupings form following the joint probability distribution of all individuals. Since this framework introduces movement, the fitness is accrued using a weighted average of payoffs at each time. However, structure in the independent movement models [41, 45] is time-independent, as in evolutionary graph theory, with interactions only depending upon a fixed structure and the population composition at the time.

The Markov model introduced in [49] (see that paper for more details of this model) and further developed here, has two further differences. One of those differences is that individuals do not occupy a fixed position, but rather move around a network based upon their previous location. Through their strategic movement they interact and accrue payoffs. The movement process considered is a variant of the *lazy random walk*, where individuals stay in the same node with a certain probability, and otherwise move to a neighbouring position chosen uniformly at random. As it will be shown, by co-evolving assortment, this is a more effective method to evolve cooperation than simple spatial correlation and the viscosity that occurs in static structured populations.

The remaining particular feature of this model is the decoupling of the payoff accrual and the evolutionary phase. Reproduction and replacement could be considered in a number of ways, but there are two natural extremes: the initial position of individuals or the positions in which their movement ends. These two distinct approaches explicitly affect our model through the replacement weights that are used within the evolutionary dynamics, since they are calculated through the proportion of time that a given pair of individuals spend together. We note that for the independent model these two concepts are equivalent, as the movement distribution is fixed through time. We consider the initial position (representing an individual's home) for its simplicity since the alternative requires individuals to never return to a home place and all information from the initial placement of individuals to be quickly lost. Thus, while any structure will have its own character in how it allows individuals to move, they would not have an intrinsic set of neighbours and, as such, locality would not be present. By considering replacement to occur in their initial positions, the structure of the population is maintained throughout the process, as after their exploration phase, the positions of individuals' are reset. This thus holds some similarities with evolutionary graph theory, keeping part of the characterising locality, which other choices wouldn't. This can be thought in terms of territorial animals who may roam great distances to find food, but return to some central nest or den to breed, as it has been observed to happen in African wild dogs [52, 53], and in migratory birds, many species of which return to the same nests every year [54].

In the present work, we propose to assess the influence of choosing different evolutionary dynamics on the interdependence between multiplayer cooperation, network topology, and assortative behaviour. We show that the evolution of cooperation is primarily dependent on network topology and that qualitative evolutionary outcomes are generally robust to the effects of distinct stochastic update rules. We evaluate this departure from previous evolutionary graph theory models [24, 46] by systematically using the set of six dynamics generalised in [46] and exploring the lasting quantitative differences observed between their results under mobile structured populations. In the following section Model, we provide an extensive description of the used framework. In Results, we develop a systematic analysis of the results obtained under complete, circle and star networks. This is done for both rare and non-rare interactive mutations and, comparing the two scenarios, we discuss the new topological effects which haven't been previously observed in [50]. Finally, in the Discussion, we summarise and analyse the similarities and differences observed between the evolutionary dynamics, in comparison with the rest of the literature.

## Model

The work accomplished in the present paper is based on the modelling framework proposed in [41]. In this section, we will introduce the general framework while focusing primarily on the aspects relevant to the Markov model considered. The framework comprises three main features which we will expand on in the following subsections: (1) network structure and Markov movement; (2) the multiplayer game; and (3) the evolutionary dynamics.

### Network structure and markov movement

Let us consider a population composed of $N$ individuals, with the $n$th individual labelled $I_n$. Each individual is positioned in a network with $M$ places, the $m$th place being labelled $P_m$. The network has a set of edges between its nodes, which will be relevant to define the possible moves individuals can make on it. We will consider the three topologies analysed in [50]: complete, star, and circle networks of different sizes. These three types of structures exhibit high degrees of symmetry, resulting in extreme clustering coefficients and degree centralisation values, both of which are critical measures in network analysis.

Although the terms "graph" and "network" are often used interchangeably in the literature, we will adopt the terminology used in [47]. Specifically, we will use the first to refer to the evolutionary graph that emerges from the replacement weights between *individuals*. On the other hand, the second will be used to refer to the network of *places* described above.

Each node in the network is home to exactly one individual, which leads to the equality $M = N$. Let $p_{n,t}(m)$ be the probability that an individual $I_n$ is at place $P_m$ at time $t$. In the context of general history-dependent movement, this probability distribution is conditional on the past positions of all individuals in the network [41], denoted as $\mathbf{M}_{t'} = [M_{n,t'}]_{n=1,\ldots,N}$, at all values of time $t' < t$. However, we are considering a Markov movement model, under which the probability is dependent only on the positions at which individuals were in the previous discrete time step, i.e. $t' = t - 1$. To make this dependence explicitly, we define the probability distribution introduced above as the following:

$$p_{n,t}(m|\boldsymbol{m}_{t-1}) = \mathbb{P}(M_{n,t} = m|\mathbf{M}_{t-1} = \boldsymbol{m}_{t-1}). \tag{1}$$

We follow the same movement model used in previous studies [49, 50]. Each individual begins the exploration phase at their home node, from which they will go through $T$ time steps, which we call the exploration time, before going back to their home nodes. At each time step $t$, individual $I_n$ evaluates the group $G_n$ in which they were at time $t - 1$. Groups are defined

as functions of the positions at the previous time step $\boldsymbol{m}_{t-1}$ and denoted as $G_n(\boldsymbol{m}_{t-1}) = \{i : m_{i,t-1} = m_{n,t-1}\}$. The probability that individual $I_n$ remains in the same place depends on their group's composition, as described by the following equation:

$$h_n(G_n(\boldsymbol{m}_{t-1})) = \frac{\alpha_n}{\alpha_n + (1 - \alpha_n)S^{\beta_{G_n(\boldsymbol{m}_{t-1})\setminus\{n\}}}}, \qquad (2)$$

where $S$ is the sensitivity, $\alpha_n$ is the staying propensity of individual $I_n$ and $\beta$ is the attractiveness of the group. The staying probability increases with the staying propensity which may hold a value between 0 and 1. Decreasing $S$ results in a greater impact of the group-dependent term on the staying probability. The attractiveness of the group with whom individual $I_n$ has interacted is obtained from the sum of the attractiveness of all other individuals in that group:

$$\beta_{G_n(\boldsymbol{m}_{t-1})\setminus\{n\}} = \sum_{i \in G_n(\boldsymbol{m}_{t-1})\setminus\{n\}} \beta_i, \qquad (3)$$

where $\beta_C = 1$ and $\beta_D = -1$ are the attractiveness of cooperators and defectors respectively.

Based on this definition, the probability $p_{n,t}(m)$ that individual $I_n$ is at place $P_m$ at time $t$ depends only on the place where the individual was in the previous step $m_{n,t-1}$, and the group they were interacting with at that time $G_n(\boldsymbol{m}_{t-1})$. This probability assumes the following form:

$$p_{n,t}(m|m_{n,t-1}, G_n(\boldsymbol{m}_{t-1})) = \begin{cases} h_n(G_n(\boldsymbol{m}_{t-1})) & m = m_{n,t-1}, \\[2mm] \dfrac{1 - h_n(G_n(\boldsymbol{m}_{t-1}))}{d(m_{n,t-1})} & m \neq m_{n,t-1} \wedge l(m, m_{t-1}) = 1, \\[2mm] 0 & l(m, m_{t-1}) > 1, \end{cases} \qquad (4)$$

where $d(m_{n,t-1})$ represents the degree of the node where $I_n$ was located at time $t - 1$, and $l(m, m_{t-1})$ represents the shortest path between the two positions in the network.

We note that in this model, if an individual decides to move, then it moves to a neighbouring location randomly with uniform probability. In other words, the only strategic decision an individual makes is whether to stay at the current location. See [55–57] for a different approach, where individuals sample all potential future locations and move strategically based on the expected payoff in each such location.

## Multiplayer game

At every time step of the exploration phase, individuals participate in a multiplayer game with the group present in the same node of the network. This interaction results in the reward $R_{n,t}$ received by individual $I_n$ at time $t$. We use the public goods game described in [49, 50], which is also known as the charitable prisoner's dilemma [40]. In this game, cooperators in a group pay a cost of $c$ to contribute $v$ to a public good, which is then equally split among all members of the group, including defectors. All individuals also receive a background reward of 1, which is intended to reduce the impact caused by extremely high selection pressure. We chose not to parameterize selection pressure explicitly since no benefit would be gained from considering the weak selection limit due to the complexity of the present model. The payoff received by the

individual in the group $G_n$ is defined as follows:

$$R_{n,t}(G_n(\boldsymbol{m}_t)) = \begin{cases} 1 - c + \dfrac{|G_n(\boldsymbol{m}_t)|_C - 1}{|G_n(\boldsymbol{m}_t)| - 1} v & \text{if } I_n \text{ is a cooperator and } |G_n(\boldsymbol{m}_t)| > 1, \\[2ex] 1 - c & \text{if } I_n \text{ is a cooperator and } |G_n(\boldsymbol{m}_t)| = 1, \\[2ex] 1 + \dfrac{|G_n(\boldsymbol{m}_t)|_C}{|G_n(\boldsymbol{m}_t)| - 1} v & \text{if } I_n \text{ is a defector and } |G_n(\boldsymbol{m}_t)| > 1, \\[2ex] 1 & \text{if } I_n \text{ is a defector and } |G_n(\boldsymbol{m}_t)| = 1, \end{cases} \tag{5}$$

where $|G_n|$ is the total number of individuals and $|G_n|_C$ the number of cooperators in the group. A distinctive feature of this multiplayer game is that the produced public good is excludable in the sense that cooperators do not benefit from their own contributions (see definition in [40]) which reduces the likelihood of cooperation evolving, similarly to what is observed in the original pairwise prisoner's dilemma. This makes it a social dilemma regardless of the values of $v$ and $c$, whereas a non-excludable version of this multiplayer game is only a social dilemma for all group sizes if $v/c < 2$ [40]. Nonetheless, the value of the reward-to-cost ratio ($v/c$) can be seen as the inverse of the dilemma strength (see [58] for an analysis of a universally scaled dilemma strength measure in pairwise games). A higher value of $v/c$, represents a lower dilemma strength and a higher social efficiency deficit [59], that is, the potential gains of moving the population from its Nash equilibrium (everyone defects) to the collective optimal state (everyone cooperates). This quantifies how easy it is for the dilemma to be solved (i.e. relaxed) through additional overlapping mechanisms [60].

At the beginning of the exploration phase ($t = 0$), all individuals start with null fitness, and the payoffs $R_{n,t}$ received at each time step $t$ will accumulate over time. The fitness contribution $f_{n,t}$ to an individual's fitness at time $t$ is calculated as follows:

$$f_{n,t}(m, G_n(\boldsymbol{m}_t)|m_{n,t-1}) = \begin{cases} R_{n,t}(G_n(\boldsymbol{m}_t)) - \lambda & m \neq m_{n,t-1}, \\[2ex] R_{n,t}(G_n(\boldsymbol{m}_t)) & m = m_{n,t-1}, \end{cases} \tag{6}$$

where $\lambda$ denotes the movement cost.

Fitness contributions are evaluated at each time step during the exploration phase until the time $T$ is reached. Consequently, the total fitness $F_{n,t}(\boldsymbol{m}_t)$ of each individual at time $t$ can be computed by summing the $T$ most recent contributions up to that time:

$$F_{n,t}(\boldsymbol{m}_t) = \sum_{t'=t-T+1}^{t} f_{n,t'}(\boldsymbol{m}_k|\boldsymbol{m}_{k-1}). \tag{7}$$

Upon completion of the exploration phase, the fitness of individuals is calculated and they are considered to go back to their respective home nodes. Thereafter, we consider an update of the population state, based on the evolutionary process described in the following subsection.

## Evolutionary dynamics

We consider the state of the population to be updated after individuals complete an exploration phase, accumulate their fitness, and return to their home nodes. During an update, one individual reproduces and another one is replaced by the first. We adopt the approach initially proposed in [23] for populations in pairwise interaction networks, and later extended in [46] for general evolutionary games on networks. We recall the definition of an evolutionary graph,

where each node represents an individual, and the adjacency matrix represents the replacement weights that determine the possible replacement events.

As suggested in [49, 50], we calculate the replacement weights based on the time individuals spend together one exploration time step after being at home. We assume that individuals spend equal fractions of time with members of the same group, and only spend time with themselves when they are alone. The time spent between two individuals $I_i$ and $I_j$ under the set of positions of the population $M = m$ is denoted $u_{i,j}$, and depends only the group $G_i(m)$ meeting with $I_i$ under those positions:

$$u_{i,j}(G_i(m)) = \begin{cases} \dfrac{1}{|G_i(m) \setminus \{i\}|} & i \neq j \wedge j \in G_i(m), \\ 0 & i \neq j \wedge j \notin G_i(m), \\ 1 & i = j \wedge |G_i| = 1, \\ 0 & i = j \wedge |G_i| > 1. \end{cases} \tag{8}$$

The replacement weights are denoted as $w_{i,j,t}$ and are considered at a time $t$ that is multiple of $T$. The positions of individuals at home are defined as $m_{t-T}$. Replacement weights correspond to the average time spent between individuals $I_i$ and $I_j$ when they are one movement time step away from home. Thus, their values can be obtained using the following equation:

$$w_{i,j,t} = \sum_m u_{i,j}(G_i(m)) p(m|m_{t-T}). \tag{9}$$

After defining the underlying evolutionary graph, we are now in a position to consider the stochastic update rules of evolutionary dynamics. The probability of selecting a specific individual and their strategy for reproduction or replacement reflects the influence of selection and the population structure. The first is included in the model by considering the fitness of individuals, and the second through their replacement weights.

Previous approaches to Markov movement models have focused on the birth-death process with selection acting during birth (BDB), as described in [45]. The state of the population is considered for updating, and the process involves two steps. First, an individual $I_i$ is selected to give birth with a probability $b_i$ proportional to their fitness. Second, an individual $I_j$ dies with a probability $d_{ij}$ proportional to the time spent with the first. The probability of individual $I_i$ replacing $I_j$ during an evolutionary update is thus given by $\tau_{ij} = b_i d_{ij}$. There are $N \times N$ possible replacement events for each population state, each of which leads to one of a limited set of possible transitions between population states.

In the present paper, we use the set of six dynamics studied in [32] and adapted in [46] to general structured population models, such as those described in [41]. These dynamics make different assumptions about the order of events and on which event selection acts. Table 1 summarises the probabilities of birth and death, or the final replacement probability, for this set of dynamics.

The first two letters of the BDB, DBD, DBB, and BDD dynamics define which event occurs first and second, while the last letter indicates on which of the events selection acts. In the DBD dynamics, selection acts during death, meaning that one individual is selected for death proportional to their inverse fitness, and one individual is selected to give birth proportional to the time spent with the first. In the DBB dynamics, selection acts on birth, meaning that an individual is selected to die randomly from the population, and one individual is selected to

**Table 1. Probabilities of birth $b_{i(j)}$ and death $d_{(i)j}$, or final probability of replacement $\tau_{ij}$, under six evolutionary dynamics.** Probabilities are indexed by the individuals $I_i$ giving birth and $I_j$ dying. When not directly provided, the replacement probability can be obtained through the product of both other probabilities.

| Dynamics | Replacement probabilities |
|---|---|
| BDB | $b_i = \dfrac{F_i}{\sum_n F_n},\, d_{ij} = \dfrac{w_{ij}}{\sum_n w_{in}}$ |
| DBD | $d_j = \dfrac{F_j^{-1}}{\sum_n F_n^{-1}},\, b_{ij} = \dfrac{w_{ij}}{\sum_n w_{nj}}$ |
| DBB | $d_j = 1/N,\, b_{ij} = \dfrac{w_{ij}F_i}{\sum_n w_{nj}F_n}$ |
| BDD | $b_i = 1/N,\, d_{ij} = \dfrac{w_{ij}F_j^{-1}}{\sum_n w_{in}F_n^{-1}}$ |
| LB | $\tau_{ij} = \dfrac{w_{ij}F_i}{\sum_{n,k} w_{nk}F_n}$ |
| LD | $\tau_{ij} = \dfrac{w_{ij}F_j^{-1}}{\sum_{n,k} w_{nk}F_k^{-1}}$ |

give birth proportional to both their fitness and the time spent with the first. The probabilities under the BDD dynamics follow the same logic presented for the previous dynamics.

Under link dynamics LB and LD, a potential replacement is chosen proportional to both the time spent between the two individuals and to, respectively, the fitness of the individual giving birth, or the inverse fitness of the individual dying. In both these dynamics, birth and death occur simultaneously and thus, replacement probabilities $\tau_{ij}$ are directly presented in Table 1.

Consider a population consisting of individuals with complex strategies that include both an interactive and a movement component. The interactive component determines whether an individual cooperates (C) or defects (D) during interactions in the public goods game presented. The movement component is defined by their staying propensity denoted as $\alpha_n$, which takes one of the values {0.01, 0.1, 0.2, ..., 0.8, 0.9, 0.99}, similarly to what was considered in previous works [49, 50]. This leads to a total of 22 possible complex strategies.

We assume the timescale in which mutations occur to be much larger than that of replacement events. Under this assumption, the evolutionary processes described above lead to dynamics of fixation, where at most two strategies are present in the population at any given time. Thus, it becomes essential to analyse fixation probability values, i.e. the probability that one individual using a mutant strategy will fixate in a population with a distinct resident strategy. However, due to the increased complexity of the used model, a proper analytical analysis becomes difficult, even if weak selection was to be considered for simplification of the process. Therefore, we have resorted to simulating a Markov process considering two mutation scenarios. In the first scenario, mutations of the interactive component of strategies are much rarer than those of the movement component. In the second scenario, mutations of both strategy components occur at the same rate.

**Rare interactive strategy mutations.** In this scenario, mutations in movement strategies occur at a higher rate than those in interactive strategies. For each interactive strategy, there exists an optimal staying propensity towards which the population evolves. The optimal staying propensity of defectors is always the maximum value of $\alpha$, which is 0.99 since there is no benefit in moving when everyone defects. As for cooperators, their value can be determined by computing fixation probabilities between all cooperator strategies. The strategy with the

optimal staying propensity cannot be invaded by any other strategy with a probability higher than the neutral fixation rate of $1/N$. Fixation probabilities can be obtained by running simulations starting with one single mutant individual subject to the evolutionary dynamics described above. The simulation ends either in a successful fixation, where all individuals use the mutant's strategy, or an unsuccessful one, where all individuals use the resident's strategy. By running this simulation for $n_t$ trials, the fixation probability can be computed from the fraction of those trials ending in successful fixations.

We assume that populations evolve towards the optimal staying propensity of their interactive strategy, after which interactive strategy mutations become relevant. We calculate the fixation probabilities of mutant cooperators against resident defectors using their optimal propensity, and consider the mutant cooperator strategy with the highest fixation probability, denoted $\rho^C$, as the fittest mutant. Similarly, we obtain the fixation probability of the fittest mutant defector against resident cooperators, denoted $\rho^D$, by performing parallel computations. We then compare these two probabilities against the neutral fixation probability of $1/N$ and classify the evolutionary outcome as one of the following:

1. If $\rho^C > 1/N$ and $\rho^D < 1/N$, then selection favours cooperation;

2. If $\rho^C < 1/N$ and $\rho^D > 1/N$, then selection favours defection;

3. If $\rho^C > 1/N$ and $\rho^D > 1/N$, then selection favours change;

4. If $\rho^C < 1/N$ and $\rho^D < 1/N$, then selection opposes change.

**Non-rare interactive strategy mutations.**   When mutations of both the interactive and movement components of strategies occur at the same timescale, a successful strategy will face individuals of both types of interactive strategies throughout the evolutionary process. Thus, the optimal staying propensities of cooperators and defectors are determined by comparing their fixation probabilities on mixed populations. We consider a mixed population of $N/2$ cooperators and $N/2$ defectors for simplicity. Cooperators will have one optimal movement strategy for each of the possible defector complex strategies, and vice versa. We define the mutually-optimal propensities as those where none of the interactive types increases their probability of fixation in the mixed population by unilaterally changing their movement strategy.

To find the pair of mutually-optimal strategies, we calculate the fixation probabilities of cooperators and defectors starting from the mixed population for all combinations of staying propensities. We find the equilibrium pair by starting with any strategy, and iterating over the fittest mutant of the opposing interactive type until we find a fixed point of the mutually-optimal pair. We classify the evolutionary outcome of the process based on the fixation probabilities, denoted $\rho_{N/2}^C$ and $\rho_{N/2}^D$, of the two equilibria strategies starting from the mixed population:

1. If $\rho_{N/2}^C > 1/2$ and $\rho_{N/2}^D < 1/2$, then selection favours cooperation;

2. If $\rho_{N/2}^C < 1/2$ and $\rho_{N/2}^D > 1/2$, then selection favours defection;

3. If $\rho_{N/2}^C \approx 1/2$ and $\rho_{N/2}^D \approx 1/2$, then selection is neutral.

## Results

In this section, we analyse the outcomes of comprehensive systematic simulations of the Markov movement model outlined earlier. Our focus is on identifying the variations caused by different evolutionary dynamics. For this purpose, we use two types of plots. The first type

illustrates the evolutionary outcomes for different value combinations of population size ($N$) and movement cost ($\lambda$). The second displays the numerical value of the fixation probabilities for cooperators and defectors as the movement cost varies.

For the plots that depict the regions where each evolutionary outcome prevails, we will employ the following colour-coding scheme:

- Blue indicates that selection favours cooperators;

- Orange indicates that selection favours defectors;

- Grey indicates that selection opposes change or is neutral;

- Yellow indicates that selection favours change.

It is important to note that the colour-coding scheme is used for both mutation scenarios, even though the non-rare interactive mutation scenario does not feature the yellow colour.

The plots with evolutionary outcomes presented here differ slightly from those in [50] for the same dynamics. In this paper, we have implemented a $2\sigma$ rule to manage stochastic uncertainty, which has been applied to both mutation scenarios. We assume that the mutant fixation probability exceeds the neutral one only if the simulated fixation probability exceeds the neutral fixation probability by at least two standard deviations. This means that for the rare interactive strategy mutations scenario the threshold is $1/N + 2\sigma$, and for the non-rare interactive mutations scenario the threshold is $1/2 + 2\sigma$. This mostly impacted the complete network as the region where selection favours defectors became slightly smaller. The estimations of the standard deviation in each case are provided in [50]. They are based on 100,000 simulation trials for each combination of parameters in the rare interactive mutations case and 10,000 simulation trials in the non-rare interactive mutations case. The thick grey lines around the neutral fixation probability value on the fixation probability plots show the $\pm 2\sigma$ area of stochastic uncertainty.

We present these plots for complete, circle, and star networks in the following sections, separated into the two scenarios of Rare interactive strategy mutations and Non-Rare interactive strategy mutations. We use the following parameter values: $S = 0:03$, $c = 0:04$, $v = 0:4$, $T = 10$. An extensive analysis of the parameter space has been done in [50], on whose S1 File the impact of different values of reward-to-cost ratio and exploration time is assessed. The two mutation scenarios in comparison shows a brief comparison of the results from the two mutation scenarios, together with some new topological effects observed in the figures throughout the Results section. However, it is in the Discussion that we do a thorough analysis, provide explanations for the similarities and differences observed between the evolutionary dynamics, and draw comparisons with previous approaches to this topic.

## Rare interactive strategy mutations

The results obtained under rare interactive strategy mutations are presented in Figs 1–6.

**Complete network.**   The evolutionary outcomes obtained under complete networks are similar for different evolutionary dynamics, as evidenced by the region plots in Fig 1. Selection promotes stability for most of the parameter space. Cooperators are favoured for lower values of the movement cost regardless of population size, while defectors do better for large movement costs and small networks.

Despite the similarities, there are still a few clear emerging differences. The region where defectors dominate is larger under dynamics where selection acts on the death event. Under the DBD and LD dynamics, selection favours defectors regardless of population size when movement costs are high enough ($\lambda \geq 0.8$), and it does so for much lower movement costs

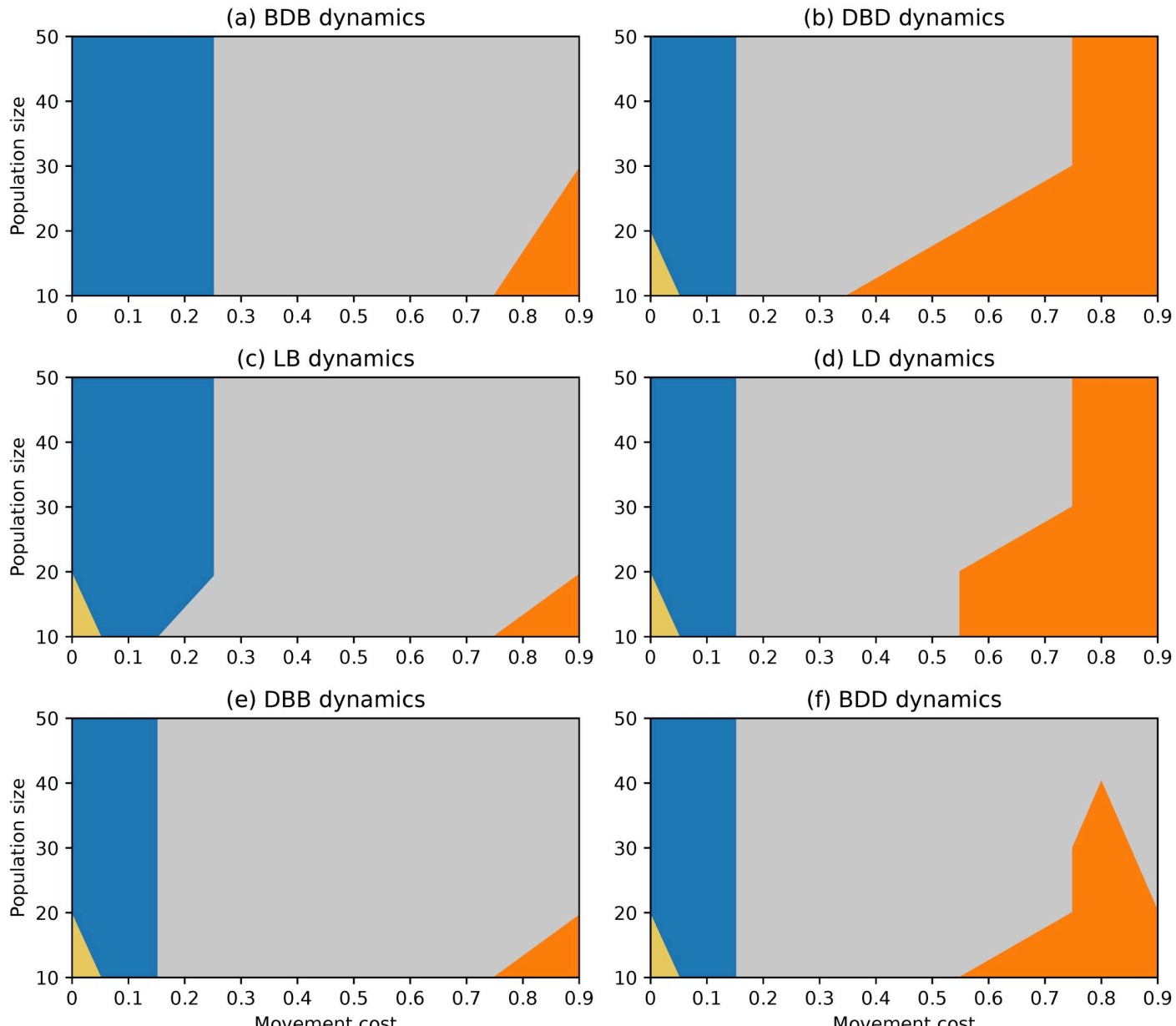

**Fig 1. Evolutionary outcomes under complete networks and rare interactive mutations for different choices of evolutionary dynamics, population size and movement cost.**

under small enough populations. The regions under which cooperation dominates are the largest under the BDB and LB dynamics. Finally, the joint stability of both strategies (selection opposing change) is favoured more often under the BDD and DBB dynamics than under the remaining dynamics.

An important aspect to highlight is that there is a small region where selection favours change under null movement costs and the smallest populations which is present under almost all dynamics. This was not documented in the analysis of the BDB dynamics on the complete network in [50], but it was observed then under the circle network.

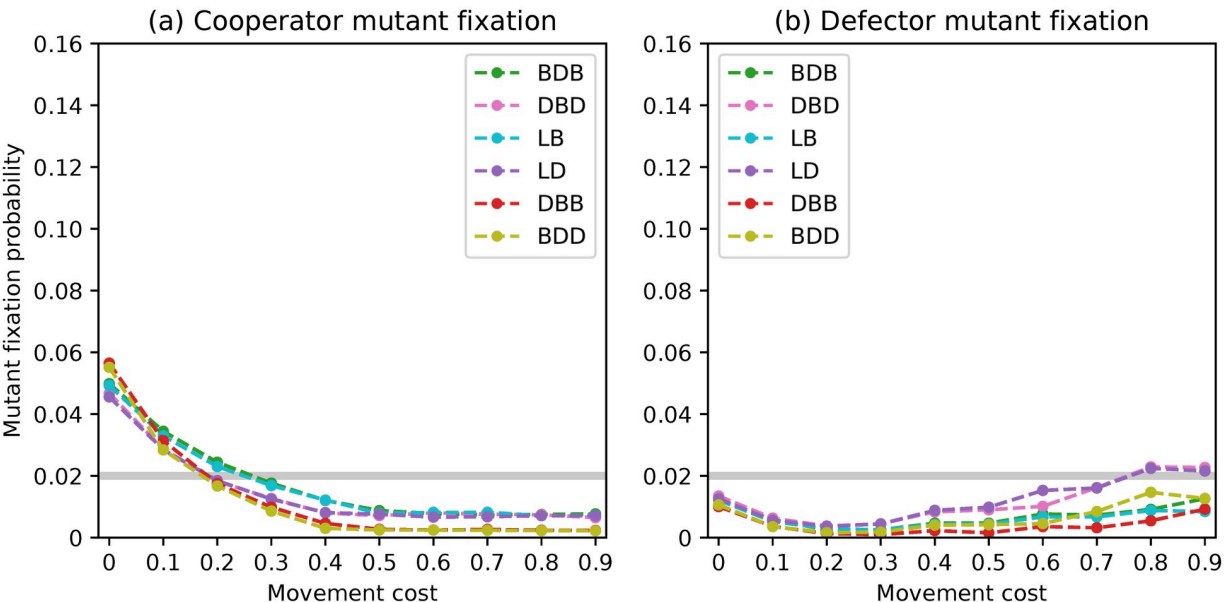

**Fig 2. Fixation probabilities of fittest mutant cooperators and defectors under a complete network with *N* = 50 and rare interactive mutations for different evolutionary dynamics.**

The fixation probabilities for *N* = 50 are displayed in Fig 2. All six evolutionary dynamics exhibit similar trends in fixation probabilities, which align with the results presented in [50], particularly for the fixation of cooperators. It is worth noting, however, that dynamics DBD and LD give defectors a chance to fixate above neutrality in populations of size 50, resulting in fixation probabilities that are twice as high as those of the other dynamics. This will be further discussed in the conclusions section.

After analysis, we discovered the formation of pairs of dynamics that led to comparable outcomes. Specifically, the BDB/LB and DBD/LD pairs had overlapping values, while the DBB/BDD dynamics had looser overlaps. This pattern emerged for both the fixation of cooperators and defectors.

Our analysis revealed the formation of pairs of dynamics leading to similar results. Specifically, the pairs BDB/LB and DBD/LD had overlapping curves and the pair DBB/BDD similarly led to quite close values. This pattern emerged for both the fixation of cooperators and defectors. There are punctual deviations observed in the first two pairs, which can be attributed to considering different discrete values for the optimal staying propensities of resident cooperators. Furthermore, we observed the overlap of the curves referring to the four dynamics BDB, LB, DBD and LD for the fixation of mutant cooperators in the presence of large movement costs, and the fixation of defectors under low movement costs. In these situations, the DBB and BDD dynamics also exhibit overlapping curves. However, differences can be observed within these pairs of dynamics for the remaining movement cost values. This result was surprising and it is discussed in the Discussion as previous work [46, 48] suggests that complete networks should lead the BDB/DBD and BDD/DBB pairs of dynamics to yield the same outcomes.

**Circle network.** The circle network leads to results presented in Fig 3. These exhibit a general pattern of single strategy stability, with cooperators dominating in regions of lower movement costs (excluding the minimal value of λ = 0) and defectors dominating in higher-cost regions.

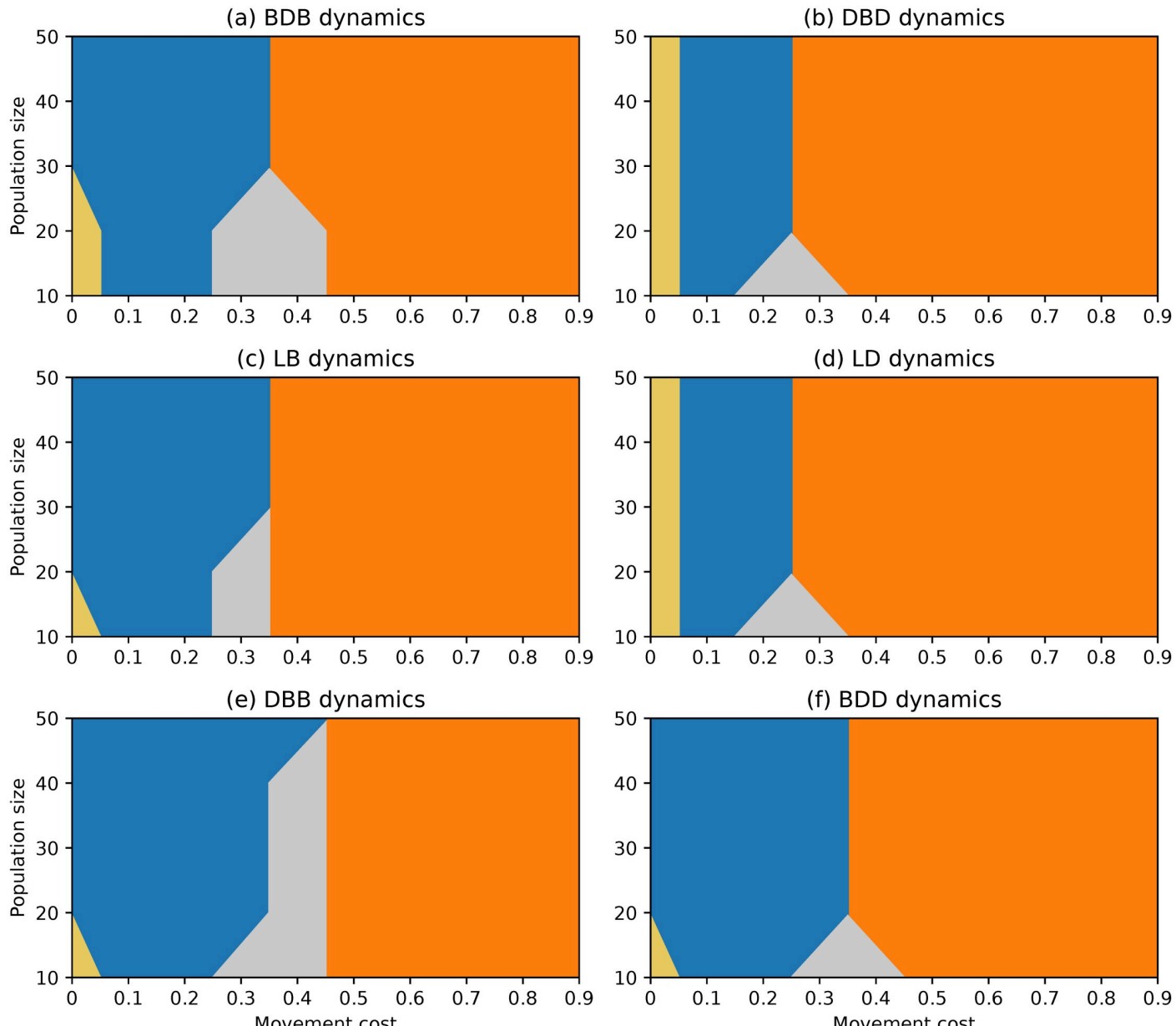

**Fig 3. Evolutionary outcomes under circle networks and rare interactive mutations for different choices of evolutionary dynamics, population size and movement cost.**

There are two exceptions to this trend: selection opposes change for intermediate values of movement costs and favours change for minimal values. The first exception typically occurs for small populations as observed in [50], which suggests it might be associated with the finiteness of populations, stabilising when these are larger. However, the second remains present regardless of population size under DBD/LD dynamics because, contrary to what happens under other dynamics, defectors fixate above $1/N$ for $\lambda = 0$ under all population sizes.

The DBD and LD dynamics exhibit unique characteristics. Compared to other dynamics, they facilitate the evolution of defection at lower values of movement costs and favour change

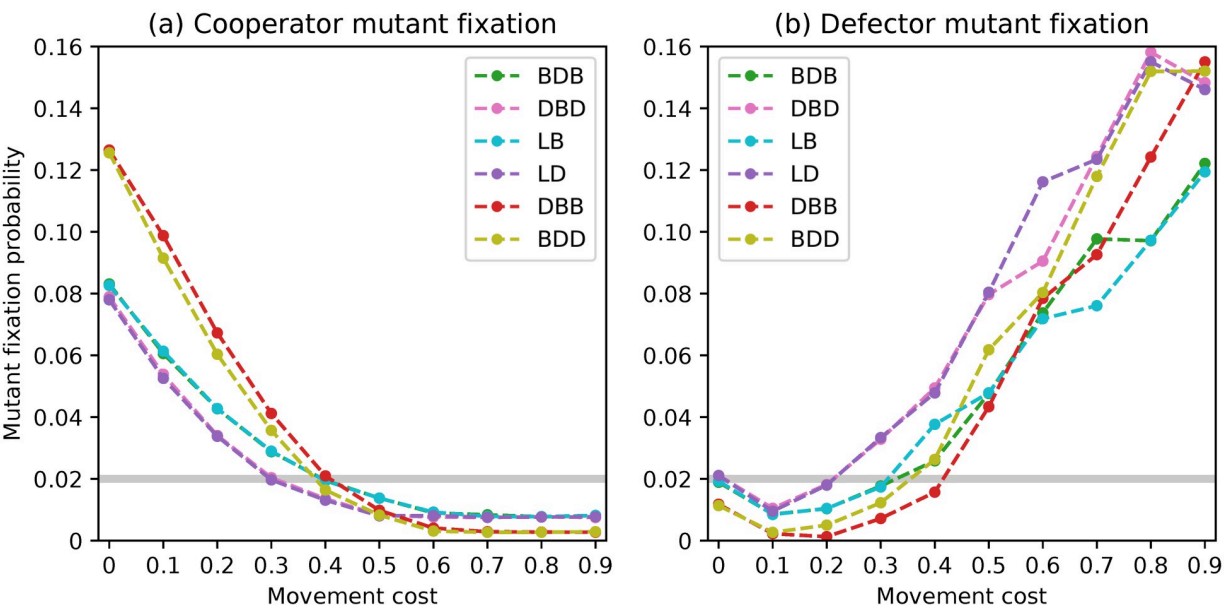

**Fig 4. Fixation probabilities of fittest mutant cooperators and defectors under a circle network with $N$ = 50 and rare interactive mutations for different evolutionary dynamics.**

for null movement costs regardless of population size. These observations suggest that these dynamics promote the fixation of defectors and hinder that of cooperators, as was seen under the complete network.

Examining the fixation probabilities displayed in Fig 4, we can easily draw the conclusion that various evolutionary dynamics follow similar patterns. The DBD/LD dynamics present smaller regions where cooperators fixate above neutrality and larger regions where defectors do so, leading to the result already displayed in Fig 3.

Similar to the complete network, the fixation probabilities of cooperators have overlapping curves within the pairs of dynamics BDB/LB and DBD/LD. The same holds true for the fixation of defectors, especially for low movement costs. However, larger costs lead to increased noise in the values, which may be linked to the fact that only discrete values of the strategic staying propensity of resident cooperators are considered. This can result in choosing either side of the scale when the optimal staying propensities fall between two discrete values. The high sensitivity of fixation probabilities to the staying propensity of residents contributes to the sudden spikes seen for $\lambda$ = 0.4, 0.7 in the BDB/LB and $\lambda$ = 0.6 in the DBD/LD pair of dynamics, in otherwise overlapping curves.

In the circle network, the BDD and DBB dynamics result in much larger deviations from neutral selection, both for the overall fixation of cooperators and for the fixation of defectors at low movement costs. The numerical difference is remarkable, with values lower than neutral achieving near-zero fixation in certain cases, and values higher than neutral being more than 50% higher than under any other dynamics.

**Star network.** The mapping of evolutionary outcomes under star networks is displayed in Fig 5. These plots exhibit minimal differences. The conclusion drawn in [50] that cooperators are consistently unstable under this topology remains valid, which is the most pronounced instance of topological effects dominating over the evolutionary dynamics.

The region plots for the four dynamics BDB, LB, DBD, and LD are identical. The high similarity within pairs BDB/LB and DBD/LD was already observed in the previous sections.

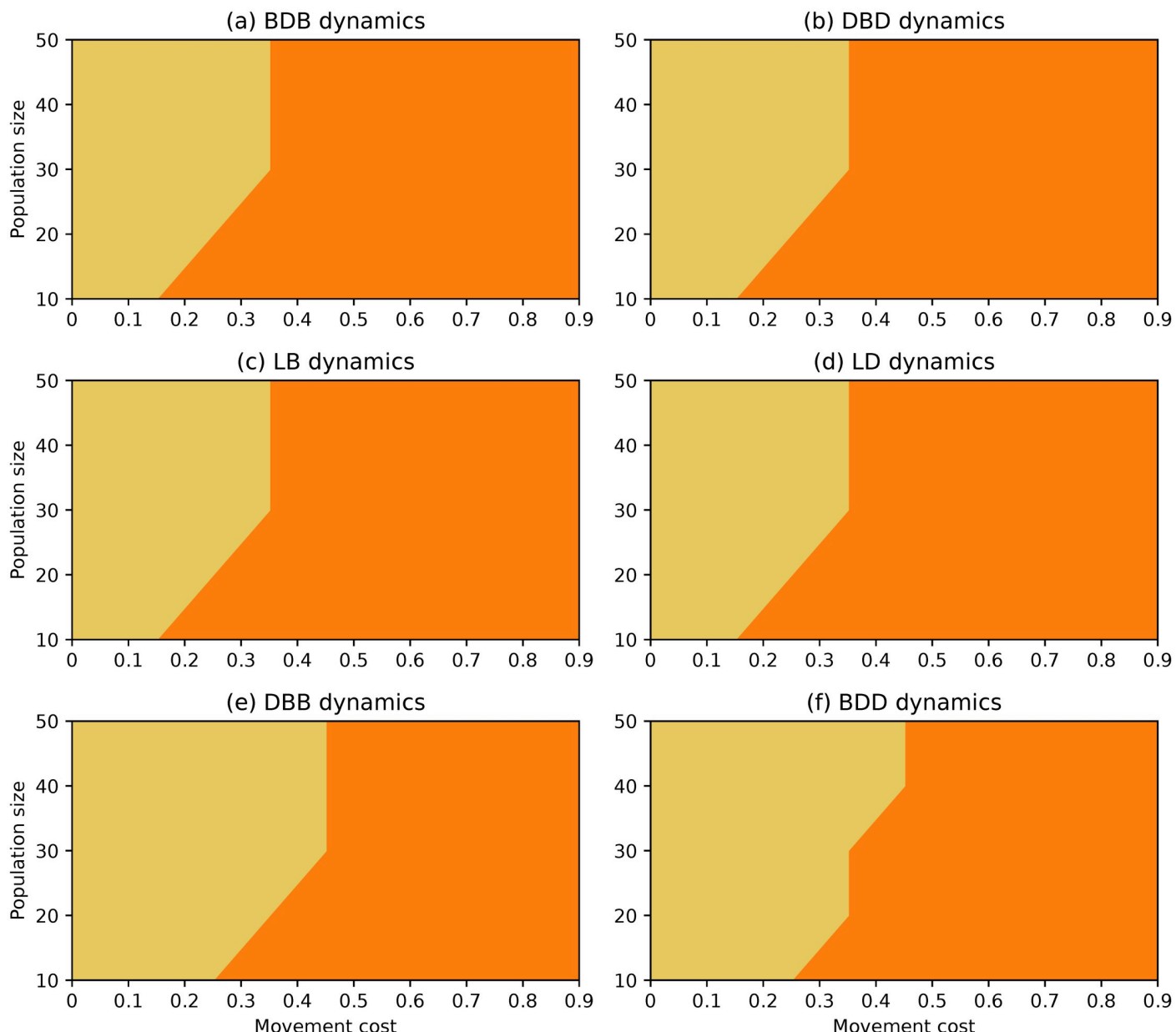

**Fig 5. Evolutionary outcomes under star networks and rare interactive mutations for different choices of evolutionary dynamics, population size and movement cost.**

However, it is surprising that the two pairs are equivalent to each other, considering the large differences observed between them in other topologies, including complete networks.

Although the differences with the two remaining dynamics DBB and BDD are minor, it is noteworthy that they appear to be more similar to each other than to the previously mentioned four other dynamics.

The fixation probabilities obtained for $N = 50$, as displayed in Fig 6, suggest a greater level of similarity among the dynamics than under other topologies. The dynamics BDB/LB and DBD/LD result in nearly identical outcomes between them for the fixation of both cooperators

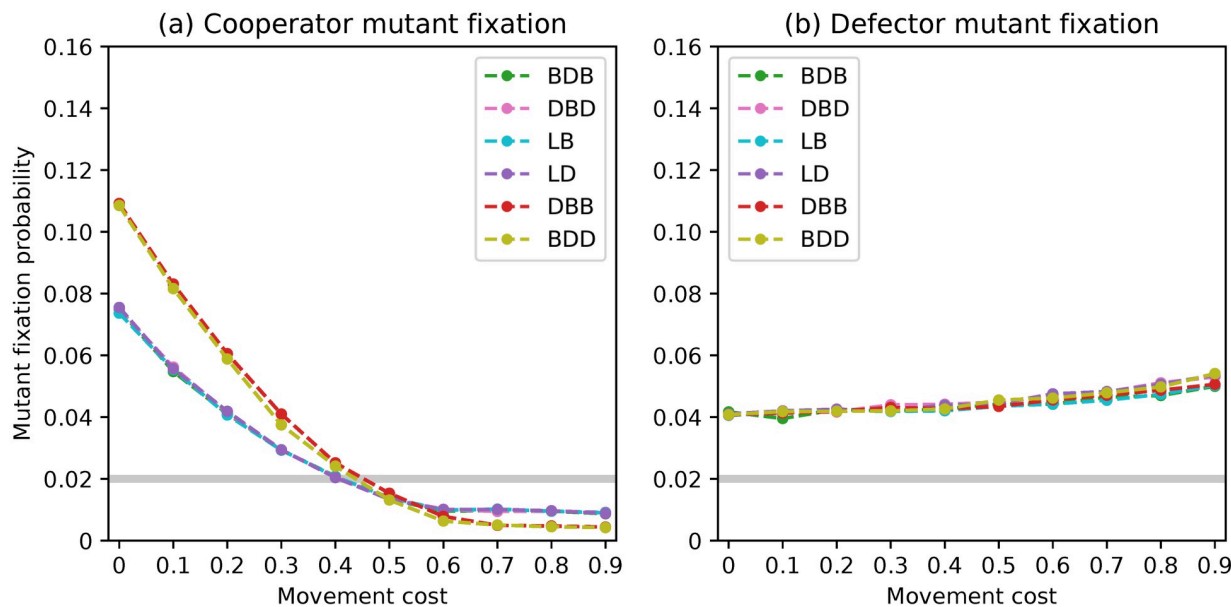

**Fig 6. Fixation probabilities of fittest mutant cooperators and defectors under a star network with *N* = 50 and rare interactive mutations for different evolutionary dynamics.**

and defectors. For the fixation of cooperators, the DBB and BDD dynamics produce results that are essentially the same between them but are systematically farther from neutrality when compared to the other four dynamics. However, for the fixation of defectors, the numerical results of all six dynamics coincide.

### Non-rare interactive strategy mutations

The results obtained under non-rare interactive strategy mutations are presented in Figs 7–12.

**Complete network.** The scenario with non-rare interactive mutations results in a significantly different landscape of evolutionary outcomes under complete networks, as depicted in Fig 7. Cooperators are favoured by selection for wide regions of low and intermediate movement costs under all dynamics. Regions in which selection does not favour either strategy are narrow and transitional, both for large movement costs and for limiting null costs. It is worth mentioning that the complete network once again appears to promote the evolution of cooperation across a broad range of parameter values.

A comparison of the results obtained under each dynamic reveals that the DBD and LD dynamics result in larger regions where defection is stable when compared to the other dynamics. Defection remains consistently stable down to λ = 0.6 regardless of population size, and for null movement costs of λ = 0 (sometimes together with cooperation) for most population sizes. In contrast, the BDB and LB dynamics remain the dynamics under which cooperation is selected across the widest regions, whereas defectors have very limited values for which they are stable.

The DBB and BDD dynamics show sets of regions somehow between the previous two pairs of dynamics. However, comparing these two dynamics, it is clear that the first is slightly more favourable towards cooperation than the second. This is akin to the previous comparison between the pairs BDB/LB and DBD/LD, suggesting the presence of a systematic difference which was not expected to be present under the complete network, which we approach in the Discussion.

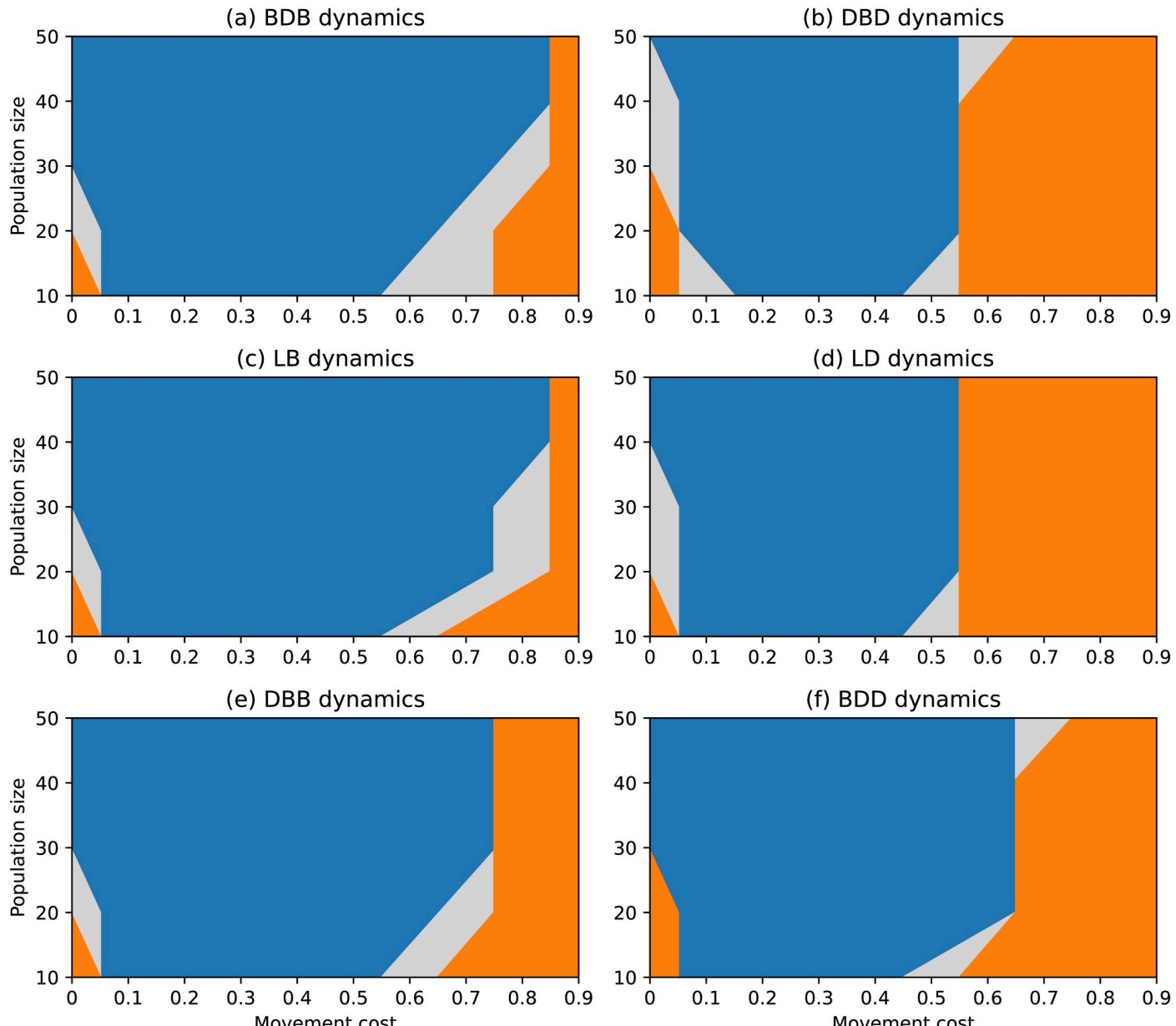

**Fig 7. Evolutionary outcomes under complete networks and non-rare interactive mutations for different choices of evolutionary dynamics, population size and movement cost.**

In this scenario, we examined the fixation probabilities starting from a mixed state with an equal number of cooperators and defectors with mutually-optimal staying propensities. As a result, the fixation probabilities of both types displayed in Fig 8 are symmetrical and sum up to one for each choice of movement cost.

Once again, the trends among the different dynamics are highly similar. The fixation probability of cooperators is at a near-neutral level for null movement costs, rising above it for intermediate costs before falling below it for higher values. Fixation probabilities are coincident within pairs BDB/LB and DBD/LD, with the first consistently leading to better outcomes for

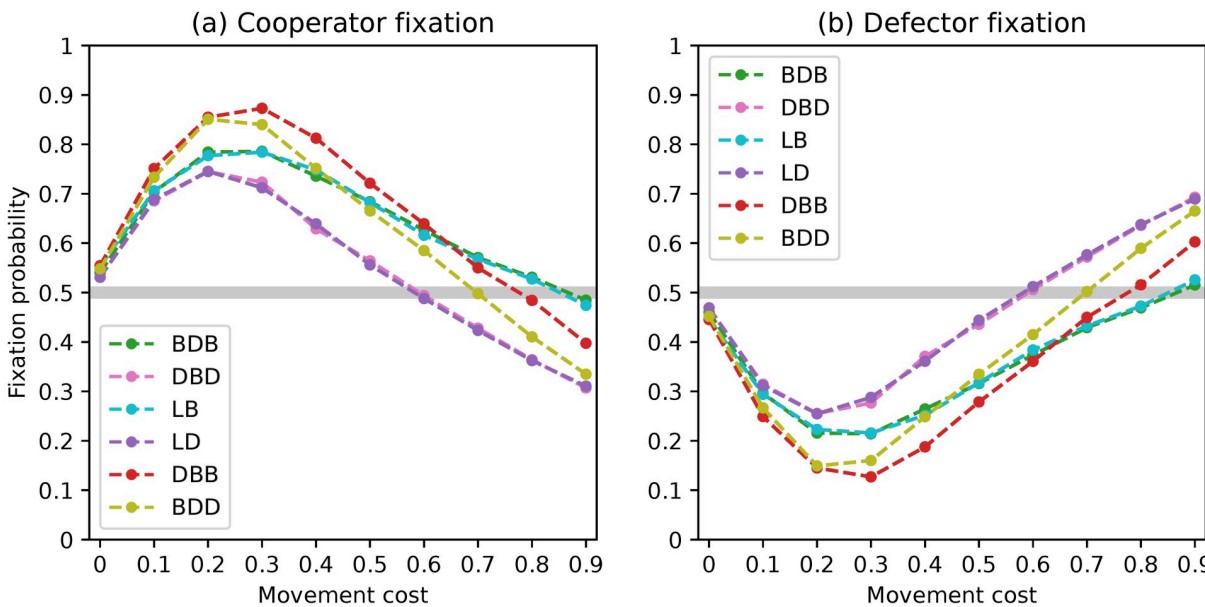

**Fig 8. Fixation probabilities of fittest mutant cooperators and defectors under a complete network with $N = 50$ and non-rare interactive mutations for different evolutionary dynamics.**

the evolution of cooperation. The DBB dynamics lead to the fixation of cooperators with higher probabilities than the BDD. These quantitative findings support the observations made from the region plots.

**Circle network.** The evolutionary outcomes obtained under circle networks with non-rare interactive mutations are exhibited in Fig 9. This figure shows that similarly to the complete network, defectors are favoured for both null and larger values of the movement cost, while cooperators are favoured from low to intermediate values of this. The transitions between cooperators to defectors showed narrow regions where selection didn't favour either of the two strategies in particular. Defectors were stable down to lower values of movement costs than under the complete network, thus assuring that, in comparison, this topology favoured them slightly more often.

The results obtained under this setting show small differences when compared to the ones obtained under rare interactive mutations for the same topology. This might be associated with the nonexistence of widespread regions where selection favours or opposes change between the two strategies under the previous mutation setting. This is a feature particular to the circle network, which is not present under the other studied topologies.

While the differences between evolutionary dynamics are not as striking as under other settings, we still observe that the DBD and LD dynamics assure the largest regions of selection of defection, for movement costs of $\lambda = 0$ and $\lambda \geq 0.4$ regardless of the size of the population.

The fixation probabilities for populations of size $N = 50$, as depicted in Fig 10, reveal certain features more clearly. The pairs of dynamics BDB/LB and DBD/LD continue to exhibit close alignment within them. The two pairs additionally converge to similar values both for low and high movement costs, values under which the DBB and BDD dynamics similarly converge to each other. For the remaining values, the BDB/LB dynamics systematically lead to higher fixation probabilities of cooperators than the DBD/LD, just like the DBB shows an improvement (even if quite small) when compared to the BDD dynamics.

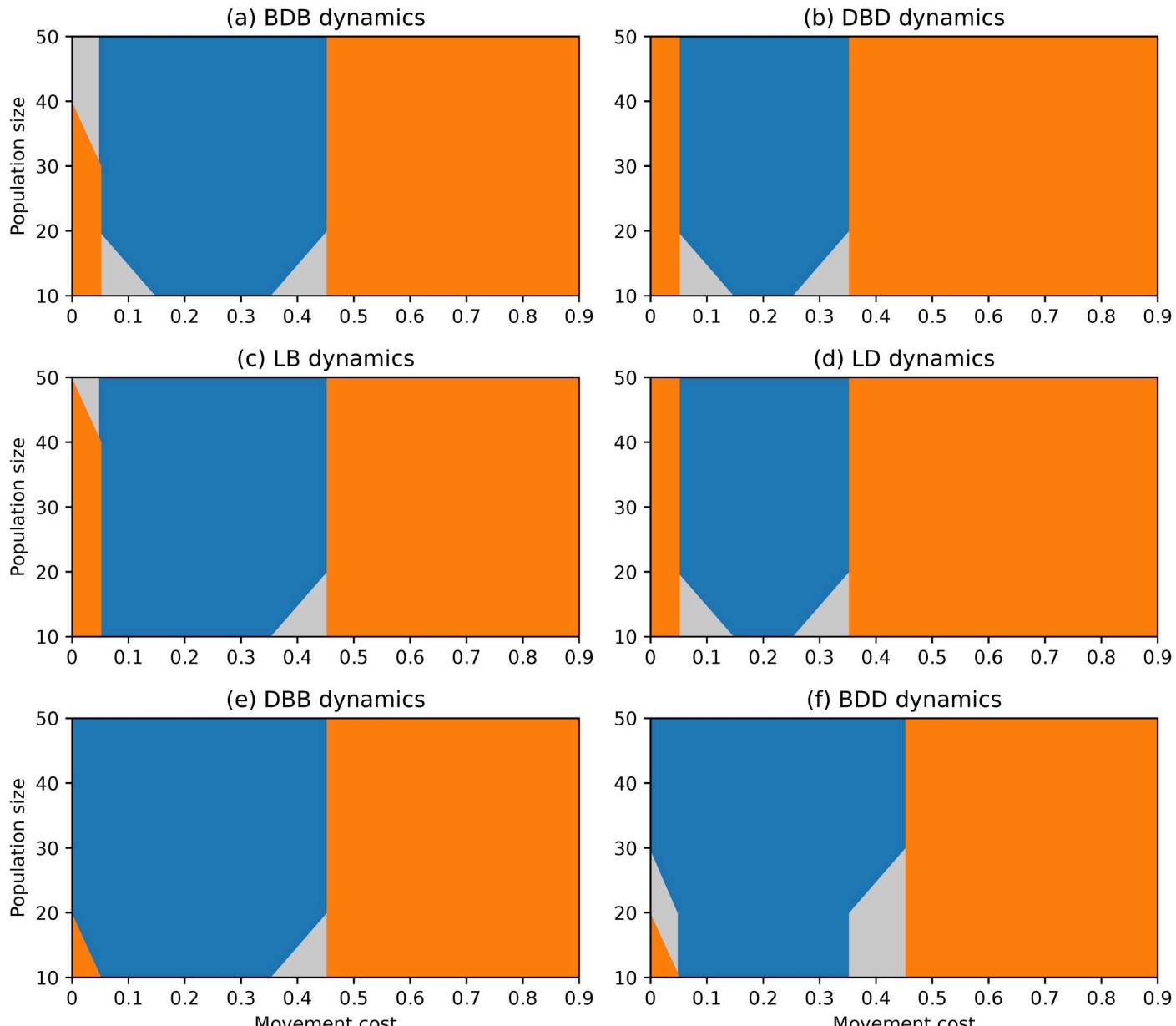

**Fig 9. Evolutionary outcomes under circle networks and non-rare interactive mutations for different choices of evolutionary dynamics, population size and movement cost.**

Finally, the DBB and BDD dynamics exhibit a clear pattern of amplified selection, with values above neutrality being the highest and values below neutrality being the lowest among all dynamics.

**Star network.** The results obtained for the evolutionary process in star networks under non-rare interactive mutations are shown in Fig 11. Unlike the results obtained in the same topology with rare interactive mutations, the different dynamics result in substantial differences in this scenario. Across all dynamics, defectors are favoured by selection at both low and high movement costs. However, there are intermediate regions where either cooperation or no

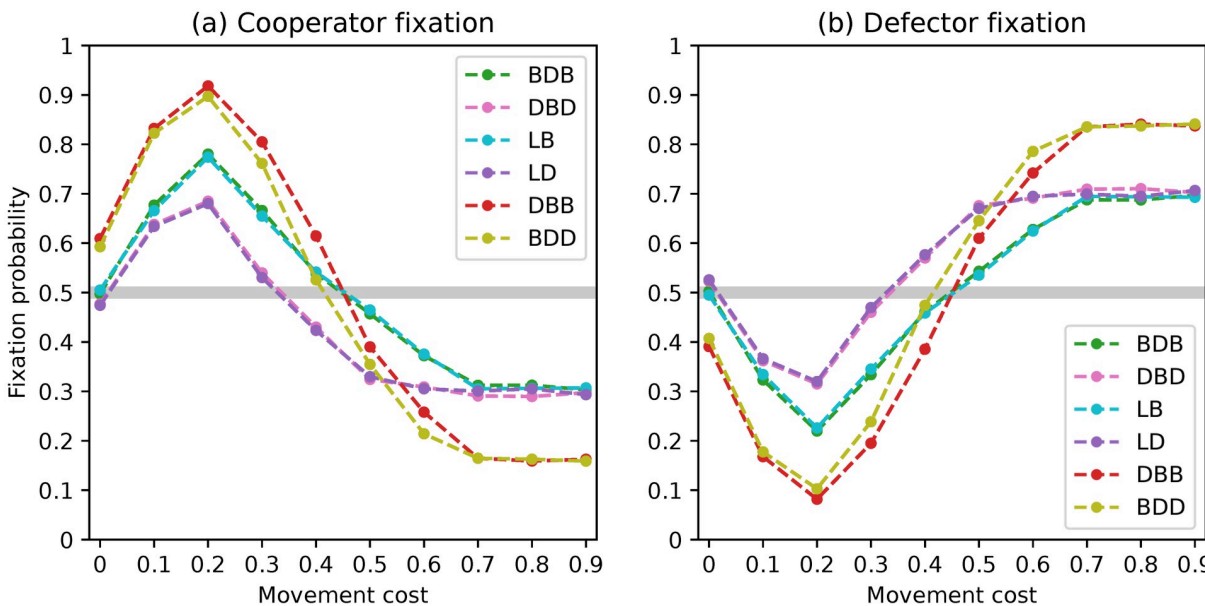

**Fig 10. Fixation probabilities of fittest mutant cooperators and defectors under a circle network with $N$ = 50 and rare interactive mutations for different evolutionary dynamics.**

strategy is favoured, and these regions are highly variable and dependent on the particular dynamics being considered.

This case presented a unique challenge that was discussed in [50]. It was usually impossible to find a mutually-optimal pair of staying propensities for cooperators and defectors. This was caused by the fact that the optimal staying propensity of defectors had a jump discontinuity as a function of the staying propensity of cooperators. We circumvented this issue in [50] by assuming that the staying propensities may change only to the nearest values. We obtained either local equilibria or local loops; in the latter case, we assumed that the optimal staying propensities corresponded to the "middle" values in the loops. We employed the same approach in this paper. This might have been the driver of the wider variation of the outcomes between different dynamics compared to the complete or circle networks.

Despite the wide variations, the BDB and LB dynamics lead to equivalent maps of evolutionary regions, under which cooperation is solidly favoured for intermediate values. Conversely, the DBD and LD dynamics exhibit larger regions of favoured defection and smaller regions of favoured cooperation, which may differ from each other due to a higher susceptibility to stochastic fluctuations. The DBB and BDD dynamics show the widest regions of favoured defection, particularly for large populations where defection is favoured regardless of the movement cost value. When comparing these two dynamics, the BDD continues to exhibit a stronger tendency to promote defectors, failing to sustain cooperation across all population sizes and movement cost values explored.

These results differ greatly from those obtained through rare interactive mutations, as this scenario does not permit selection to favour change. In the previous mutation scenario, not only did we observe little to no differences between dynamics, but we also observed cooperation to be unstable for all explored values. Therefore, this mutation scenario presents an opportunity for cooperators to evolve within star networks.

Fig 12 displays the fixation probabilities of cooperators and defectors when $N$ = 50. Fixation probability values obtained under different dynamics are quite similar quantitatively.

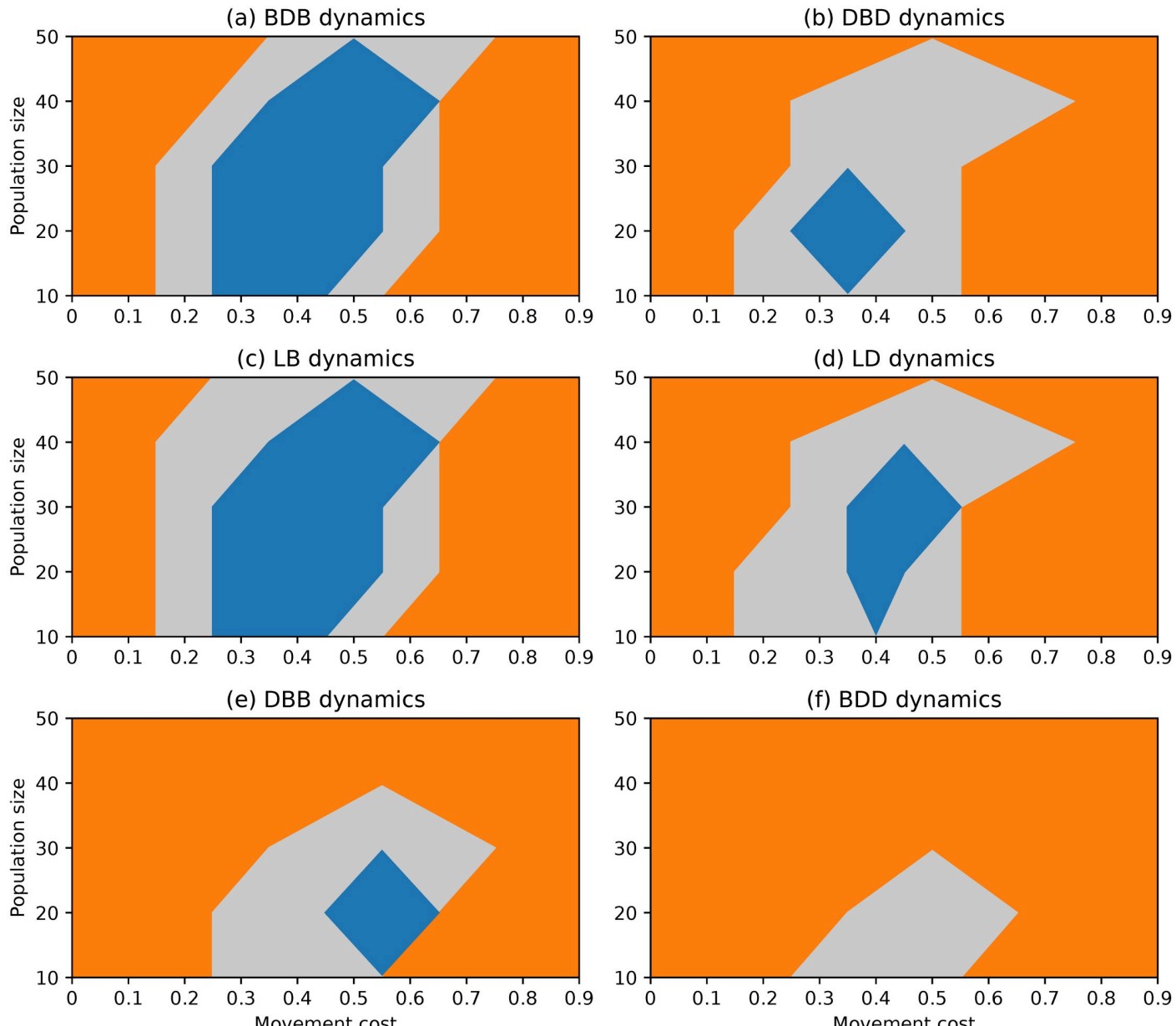

**Fig 11. Evolutionary outcomes under star networks and non-rare interactive mutations for different choices of evolutionary dynamics, population size and movement cost.**

However, the proximity to the neutral selection fixation probability of 1/2 allows for small differences between the dynamics to potentiate distinct qualitative evolutionary outcomes.

It is still clear that the BDD and DBB dynamics hold fixation probabilities the furthest away from neutrality, in this case promoting more often than other dynamics the evolution of defection. Selection happening on the birth event still seems to benefit the fixation of cooperators and oppose that of defectors slightly, which can be seen by comparing the BDB/LB dynamics against the DBD/LD and the DBB dynamics against the BDD.

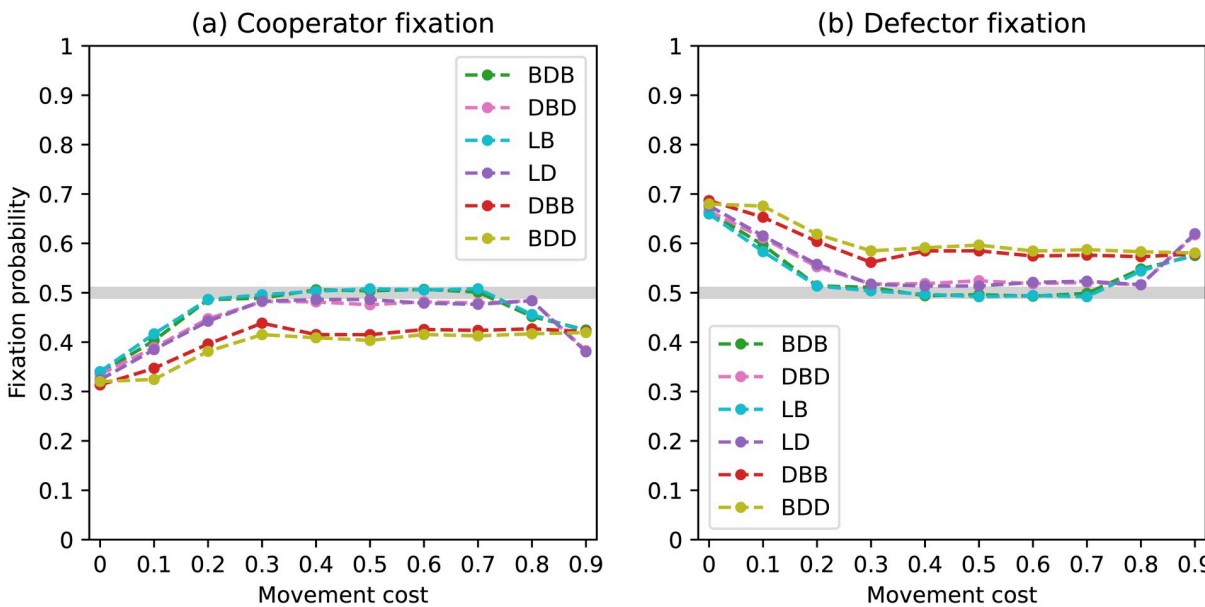

**Fig 12. Fixation probabilities of fittest mutant cooperators and defectors under a star network with *N* = 50 and rare interactive mutations for different evolutionary dynamics.**

## The two mutation scenarios in comparison

The two mutation scenarios resulted in distinct evolutionary outcomes, partially due to their different nature. In the first, selection favouring change was consistently observed in the complete and circle networks under null movement costs, and in the star network under null-to-intermediate movement costs. The frequent presence of these regions was brought to light through the examination of alternative evolutionary dynamics to the BDB used in [50], since those uncovered the region in the complete network and extended it for larger populations in the circle network.

In that scenario, mutant cooperators systematically fixate above neutrality for low enough movement costs and for all topologies and evolutionary dynamics. Selection favouring change for $\lambda = 0$ is thus associated with mutant defectors doing so as well in that limit. The stability of resident cooperators relies on the extra steps that defectors have to do before finding groups of cooperators to exploit. Even though defectors benefit from moving (shown by their fittest mutant's staying propensity not being 0.99), they still earn less than cooperators because of the higher movement cost they pay and the limited time they spend amongst them. When $\lambda = 0$, the absence of movement costs gives defectors an evolutionary advantage, leading to a fixation above neutrality. This occurs regardless of population size in the circle network for some dynamics, possibly due to its locality being preserved under larger populations, enabling defectors to quickly encounter groups of cooperators.

Additionally, the first mutation scenario led to significantly noisier fixation probabilities of mutant defectors and less distinguishable patterns. This was due to optimal staying propensities of resident cooperators being dependent on the movement cost—contrary to the constant staying propensity of 0.99 achieved by resident defectors—which had to be calculated from a discrete set of values. This computation added uncertainty to the resulting fixation probabilities which made the distinction of patterns comparatively more difficult, especially for larger movement costs.

Upon examination of the results obtained from the non-rare interactive mutation scenario and comparison with the previous, it becomes evident that different topologies result in distinct relationships. We observe that regions where selection favours one single strategy (i.e. cooperators or defectors) in the first mutation scenario typically carry over into the second scenario. However, regions under which selection favours/opposes change can fall onto any of the possible evolutionary outcomes in the second scenario. These shifts are especially prominent in topologies such as the complete network where selection opposing change is a prevalent outcome, or the star network where selection often favours change.

The distinctive nature of the star network is once again evident in the substantial variability of evolutionary outcomes observed in the non-rare interactive mutation scenario. This is attributed to the proximity of fixation probabilities to the neutral fixation value of 1/2, which results in small quantitative changes having a significant impact on the qualitative outcomes. This phenomenon may be linked to the jump discontinuity reported in [50] and mentioned in the context of the results obtained under Non-Rare interactive strategy mutations. This jump occurs in the mutually-optimal staying propensities, potentially leading to the dynamics taking on a decisive role as it is observed in Figs 11 and 12. We see instances where cooperation evolves in regions where defectors consistently dominated under rare interactive mutations.

Furthermore, we have observed both surprising similarities and novel differences between the outcomes produced by different evolutionary dynamics, some of which emerged systematically across various topologies and mutation scenarios. We summarise and analyse them in the final section of this paper in comparison to what the previous literature has suggested to us. We anticipate their influence to extend beyond the scope of the specific population structure and mobility model utilised in this study.

## Discussion

We present a comprehensive analysis of the variations obtained between six distinct evolutionary dynamics on the evolution of cooperation within structured populations following Markov movement. These dynamics have been studied on graphs in works such as [23, 24, 29, 32]. They were generalised in [46] to allow their usage in a broader range of structured population models, such as the ones used there and in [48]. Our examination of these dynamics under the three extreme network topologies studied in [50] brought to light several key aspects.

The most striking feature is that the set of evolutionary dynamics analysed yields overall qualitatively similar results, indicating that network topology has a greater influence than the particular dynamics considered. The features that characterise evolutionary outcomes under each topology, some of which were already pointed out in [50], are shown to hold across evolutionary dynamics. A deviation from this pattern was observed in the star network with non-rare interactive mutations, a scenario that was highlighted in both this study and in [50] for its unique properties.

However, the pervasive similarity of qualitative outcomes obtained under *all* dynamics came out as a surprising result, considering the substantial differences that some dynamics have shown in promoting cooperative behaviour in the past. In the groundbreaking paper [24], it is shown that the DBB dynamics (and the BDD, by extension) reveal the viscosity of evolutionary processes on networks, thus leading to the evolution of cooperation without the need for other overlapping mechanisms to be present. In summary, cooperation evolved in those networks if the reward-to-cost ratio surpassed the average number of neighbours each individual had, $v/c > \langle k \rangle$, and the network was far from complete, $\langle k \rangle \ll N$. This was presented in stark contrast to the results obtained under the BDB dynamics (and DBD, by extension), under which viscosity is not reflected and therefore network structure alone was not sufficient

for cooperation to evolve. This was similarly observed under the territorial model where individuals played a multiplayer charitable prisoner's dilemma in a network [46]. In both of these models, replacement events and the interactions between individuals are characterized by their locality. The first assures that, compared to defectors, cooperators are more often surrounded by other cooperators, while the second guarantees that this generates an evolutionary advantage to cooperate if rewards are high enough.

The present Markov model presents a distinct picture from previous models. Although replacement events maintain their locality similarly to what was considered in [50] (see Model), considering an exploration time of $T = 10$ enables individuals to navigate the network contingent on whom they meet. This leads to two important consequences. On the one hand, it partially suppresses the impact of structural viscosity, which is heavily dynamic-dependent and the only effect present in [24, 46]. On the other hand, the assortative behaviour which co-evolves under complete and circle networks proves to be much more powerful than viscosity in promoting cooperation and succeeds in doing so under all evolutionary dynamics for low enough movement costs, and large enough reward-to-cost ratios (i.e. a lower dilemma strength [60]). Alternative strategy-dependent mobility and linking rules (and the resulting assortative behaviour) have been shown to have a parallel positive impact, not only in other models of cooperation emergence [61, 62] (see [63] for a review on co-evolving linking and mobile rules), but also in decreasing disease spreading [64, 65].

The lower viscosity together with the wide-spread presence of assortative behaviour explains why the DBB (and BDD) dynamics are not exceptional in ensuring the evolution of cooperation, as they were in [24, 46]. As a result of this, considering lower exploration times in Markov models strongly hinders the evolution of cooperation as it is shown in the S1 File from [50]. To further prove this point, it is shown in the S1 File of the present paper, that under $T = 1$ (when assortative behaviour is no longer present), the differences between the dynamics reemerge, particularly under the circle network, and when the dilemma strength is low enough and the dilemma becomes easier to relax through additional overlapping mechanisms [58]. In that case, the DBB dynamics evolve cooperation successfully, contrary to the BDB dynamics.

Moving beyond the qualitative resemblance of the evolutionary outcomes, we have observed that the two pairs of dynamics BDB/LB and DBD/LD were quantitatively equivalent within each pair. Their equivalence stems from the general underlying framework of multiplayer games in networks [41], under which the evolutionary graph is calculated from the time any two individuals spend together, with time spent alone included as a self-replacement weight. The total time passed is the same for each individual, thus resulting in an isothermal graph and the reported equivalent pairs of dynamics [29, 46].

Moreover, the DBB and BDD pair of dynamics, and to a lesser extent the BDB and DBD pair, sometimes showed similar values. The statistical study performed in [48] concluded that the dynamics within each of these pairs may result in equivalent fixation probability distributions under independent movement. While both pairs passed this test, the first pair exhibited a closer affinity than the second, a characteristic that appears to have been carried over into the results we obtained under a more complex Markov movement model.

Nonetheless, there were a few differences persisting between the evolutionary outcomes under the six dynamics. The BDB/LB dynamics were found to promote the evolution of cooperation over a wider range of parameter values, while the DBD/LD dynamics did the same for the evolution of defection. A systematic comparison between the two pairs of dynamics revealed that cooperators had higher fixation probabilities in the first pair, while defectors had higher fixation probabilities in the second. This pattern held across all topologies and mutation scenarios, with only rare and isolated exceptions.

Although the difference was more pronounced when comparing those pairs of dynamics, it was also present between the DBB and BDD dynamics. Together with the previous observation, this suggests that cooperation is overall favoured by selection when this acts during birth rather than death, regardless of whether this is the first or the second, or indeed referring to simultaneous events. According to previous results [46], the replacement structure being symmetric and doubly stochastic should result in equivalent pairs of dynamics. However, this is only true when choosing a different replacement pair between the same types does not change the future fitness of individuals [48], such as under complete networks or fixed fitness. In our results, fitness being highly variable surprisingly leads to the consistently reported differences within pairs of dynamics BDB/DBD, DBB/BDD and LB/LD, even under complete networks, and this is the subject of a forthcoming paper.

Another distinction between the dynamics is that in the second mutation case and for the fixation of cooperators in the first mutation case, the DBB and BDD consistently amplify selection compared to the other dynamics across topologies. This effect has been noted under the territorial raider model and it is analysed in a paper, which will be submitted soon. When selection acts during the second event, fitness and replacement weights become intertwined in the same probability. The replacement structure is often biased towards individuals of the same type, for example, when individuals spend a disproportionate fraction of time alone. In those cases, the DBB and BDD dynamics systematically favour the replacement of individuals with lower fitness by ones with higher fitness, thus acting as amplifiers of selection, when compared to dynamics where fitness and replacement structure are considered separately.

The only setting where the amplification effect was less pervasive was the fixation of defectors in the non-rare interactive mutation case. This is potentially associated with both mutants and residents having low staying propensities, leading to lower self-replacement weights and, therefore, a lesser bias towards same-type replacement. The star network serves as a notable limiting case, where both the mutant defector's and resident cooperator's staying propensity is 0.01, and under which defector fixation probabilities are the same for all dynamics, thus showing no amplification by these dynamics.

Due to the computational complexity of the current model, we considered only the extreme network topologies here using the complete, circle, and star networks. We looked at a wider range of network topologies in [66] (for the BDB updating mechanism) and discovered that network topology affects the evolutionary outcomes only for networks of small average degree. Once the average degree becomes sufficiently high, the outcomes match those for the complete graph. However, the actual value of the average degree when this happens is much lower than that of the complete graph.

Further investigations on evolutionary models of finite structured populations could focus on the interplay between structure and assortative behaviour in promoting cooperation. The results obtained using this particular model highlight broader features of models incorporating these aspects and should be taken into account accordingly. Nevertheless, the model of population structure and mobility introduced in [41] shows once again its flexibility. It offers the ability to create new theoretical tools and study specific evolutionary systems. The model is well-suited for analysing aggressive behaviour in territorial patches of biological populations. Additionally, it has potential in social sciences, such as in the study of labour market dynamics where employer networks could be viewed as territorial networks through which individuals move. It is our hope that this original modelling framework and all the advancements made thus far will provide valuable insights into real-world systems like these.

## Supporting information

**S1 File. Supplementary material.** Presentation and analysis of the data obtained with exploration phase length set at $T = 1$, under the two dynamics BDB and DBB, and different values of reward $c$.
(PDF)

## Author Contributions

**Conceptualization:** Igor V. Erovenko, Mark Broom.

**Formal analysis:** Diogo L. Pires, Igor V. Erovenko.

**Funding acquisition:** Mark Broom.

**Methodology:** Diogo L. Pires, Igor V. Erovenko, Mark Broom.

**Software:** Diogo L. Pires, Igor V. Erovenko.

**Visualization:** Diogo L. Pires, Igor V. Erovenko.

**Writing – original draft:** Diogo L. Pires, Igor V. Erovenko, Mark Broom.

**Writing – review & editing:** Diogo L. Pires, Igor V. Erovenko, Mark Broom.

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
