## [Decision Letter · Decision Letter 0]

16 May 2023

PONE-D-23-12612Network topology and movement cost, not updating mechanism, determine the evolution of cooperation in mobile structured populationsPLOS ONE

Dear Dr. Broom,

Thank you for submitting your manuscript to PLOS ONE. After careful consideration, we feel that it has merit but does not fully meet PLOS ONE’s publication criteria as it currently stands. Therefore, we invite you to submit a revised version of the manuscript that addresses the points raised during the review process.

We look forward to receiving your revised manuscript.

Kind regards,

Jun Tanimoto

Academic Editor

PLOS ONE

Journal Requirements:

3. We note that you have stated that you will provide repository information for your data at acceptance. Should your manuscript be accepted for publication, we will hold it until you provide the relevant accession numbers or DOIs necessary to access your data. If you wish to make changes to your Data Availability statement, please describe these changes in your cover letter and we will update your Data Availability statement to reflect the information you provide

Reviewers' comments:

Reviewer's Responses to Questions

**Comments to the Author**

1. Is the manuscript technically sound, and do the data support the conclusions?

Reviewer #1: Yes

Reviewer #2: Yes

2. Has the statistical analysis been performed appropriately and rigorously? 

Reviewer #1: Yes

Reviewer #2: Yes

3. Have the authors made all data underlying the findings in their manuscript fully available?

Reviewer #1: Yes

Reviewer #2: Yes

4. Is the manuscript presented in an intelligible fashion and written in standard English?

Reviewer #1: Yes

Reviewer #2: Yes

5. Review Comments to the Author

Reviewer #1: This establishes a new network variant-PGG model where an agent is able to mobile on the underlying network. In line with theoretical treatment, the authors carefully introduce the network structure as a Markov movement, which seems novel and interesting in terms of theoretical robustness. Agent-n’s action of whether staying on his present site (node) or mobile to one of vacant site (because they assumed M = N) in his direct neighborhood, randomly selected, is stipulated by h_n(G_n(m_t-1)); Eq. (2), which accounts both his staying propensity (denoted by Alpha_n) and the accumulated attractiveness of the group Agent-n belonging at time-step; t-1 (evaluated by Eq. (3)). Whenever an agent moves, he must pay the mobile cost; Lambda; Eq. (6).

Agent-n plays a modified PGG with his neighborhood; G_n(m_t)), where the cooperation cost; c, amplification factor (dilemma weakness; in other words, v), and the baseline income; 1, are implemented; Eq. (5).

In the exploration phase until the time T is reached, each agent accumulates fitness; f, to the time-accumulated one; F.

After the exploration phase, an evolutionary phase takes place, where one of the several variants of DB and BD processes occurs, in which the probability of Agent-I_i being selected to give birth; b_i, the probability of I_i replacing Agent-I_j; d_ij, are evaluated. An event of I_i replacing I_j is dependent on the time I_i neighboring with I_j in the exploration phase, quantified by Eq. (9). Subsequently, the authors developed six different dynamical processes as listed in Table 1.

Basically, the authors were concerned on the fixation probability; Rho^(C) and Rho^(D) that are compared with 1/N, when varying the mobile cost; Lambda, the population size; N besides the evolutionary dynamics (six update rules as above). And, the authors analyze this framework for different underlying network topology; complete graph, circle, and star graph.

Visual results they delivered are phase diagram along Lambda and N, line-graph of Rho^(C) and Rho^(D) along Lambda.

They insisted: that the cooperation is primarily dependent upon the network topology and movement cost while using diﬀerent stochastic update rules seldom inﬂuences evolutionary outcomes. Cooperation robustly co-evolves with movement on complete networks and structure has a partially detrimental eﬀect on it, which is very much contrasting to the precursors’ results that cooperation can only emerge under some update rules and if the average degree is low. The authors claimed that group-dependent movement erases the locality of interactions, suppresses the impact of evolutionary structural viscosity on the fitness of individuals, and leads to assortative behavior that is much more powerful than viscosity in promoting cooperation.

Certainly, what the authors insisting seems novel and interesting. And the approach they took is scientifically robust and reliable. Hence, I have a quite positive impression from their work. Yet, I would like to give some inquiries as below so as to bolster the impressiveness by this nice work.

#1.

The model is nice because the framework is generally described that is suitable for a theoretical analysis. But, I see their model extremely specific than a simple spatial version of PGG with a mobile option. For instance, what they introduced for the exploration phase and evolutionary one seems quite specific, neither common with the previous models as above nor generally observed in a real society at all. I do believe that the authors need more deliberate explanation of why their model can be justified.

#2.

Following to the previous item; #1, I thing that the statement; their finding is very much contrasting to the precursors’ results that cooperation can only emerge under some update rules and if the average degree is low, should be more deliberately proved. I totally agree that this is the most important finding in the present study, which (literally) attracts many audience in the arena. But I’m skeptical whether this statement was fairly proved or not. It’s because the most of the previous studies in view of a spatial version PGG with a mobile option are different from theirs that can be said too specific as abovementioned.

#3.

The authors mainly varied the mobile cost, the population besides the evolutionary dynamics and network topology. That’s fine. But everyone unequivocally perceives that there is another important model parameter of which sensitivity should be carefully explored. That is the dilemma strength of PGG. In the present model, the dilemma strength is influenced by v, c and the baseline income; +1. By referring to the universal dilemma strength; c/v relates to the dilemma strength, while v/c (for which, in the conventional PGG, b/c is adopted) indicates the dilemma weakness. I guess that they did fix the dilemma strength (dilemma weakness). They need further discussion on this point with sharing another series of results. When they add this, they should reference to the universal concept of dilemma strength for both 2-player & 2-strategy games (including PD, Snowdrift, Stag Hunt etc) and multi-player & 2-staretgy including PGG with citation of the relevant literature; for instance, (i) Social efficiency deficit deciphers social dilemmas, Scientific Reports 10, 16092, 2020, (ii) Sociophysics Approach to Epidemics, Springer, 2021.

Reviewer #2: The authors studied the evolution of multiplayer cooperation in mobile structured populations, where individuals move strategically on networks and interact with those they meet in groups of variable size. They find that the evolution of multiplayer cooperation is primarily dependent upon the network topology and movement cost while using different stochastic update rules seldom influences evolutionary outcomes. These findings contrast an established wisdom in evolutionary graph theory that cooperation can only emerge under some update rules and if the average degree is low. After I read the manuscript, I have several questions and listed as follows:

1. In the manuscript, the definition of fitness in equation 7 and the game dynamics listed in Table 1 are shown to be independent of the selection strength, indicating that the results were obtained under strong selection scenarios. However, it is worth considering the potential impact of weak selection on the evolutionary outcomes. Many existing studies have highlighted the significance of weak selection in shaping evolutionary dynamics. Therefore, I would like to inquire about the robustness of the presented results with respect to variations in the selection strength.

2. The authors chose three types of networks, namely the complete graph, circle network, and star network. However, it would be helpful to clarify whether these networks are representative in the current study. In the literature, it is widely recognized that cooperators can thrive in social networks due to network reciprocity, where cooperators tend to form cohesive clusters to support one another. Therefore, I suggest considering the inclusion of a regular lattice as it is often considered more representative. Additionally, by adjusting the short-cutting probability of links in a square lattice with a degree of 8, it is possible to generate various homogeneous networks with different clustering coefficients.

3. Section of Discussion. The discussion section appears to be relatively brief and lacks a thorough comparison with existing studies that argue cooperation can only emerge under specific update rules and when the average degree is low. To enhance the discussion, I suggest providing a more comprehensive analysis and explaining the divergence between the findings presented in this study and the existing literature. This will help readers understand the novelty and significance of the results obtained.

6. PLOS authors have the option to publish the peer review history of their article (what does this mean?). If published, this will include your full peer review and any attached files.

Reviewer #1: No

Reviewer #2: No

---

## [Author Response · Author response to Decision Letter 0]

7 Jul 2023

We have attached a detailed "Response to Reviewers" document as a separate attachment within the files.

---

## [Decision Letter · Decision Letter 1]

18 Jul 2023

Network topology and movement cost, not updating mechanism, determine the evolution of cooperation in mobile structured populations

PONE-D-23-12612R1

Dear Dr. Broom,

We’re pleased to inform you that your manuscript has been judged scientifically suitable for publication and will be formally accepted for publication once it meets all outstanding technical requirements.

Kind regards,

Jun Tanimoto

Academic Editor

PLOS ONE

Additional Editor Comments (optional):

Reviewers' comments:

Reviewer's Responses to Questions

**Comments to the Author**

1. If the authors have adequately addressed your comments raised in a previous round of review and you feel that this manuscript is now acceptable for publication, you may indicate that here to bypass the “Comments to the Author” section, enter your conflict of interest statement in the “Confidential to Editor” section, and submit your "Accept" recommendation.

Reviewer #1: All comments have been addressed

Reviewer #2: (No Response)

2. Is the manuscript technically sound, and do the data support the conclusions?

Reviewer #1: Yes

Reviewer #2: (No Response)

3. Has the statistical analysis been performed appropriately and rigorously? 

Reviewer #1: Yes

Reviewer #2: (No Response)

4. Have the authors made all data underlying the findings in their manuscript fully available?

Reviewer #1: Yes

Reviewer #2: (No Response)

5. Is the manuscript presented in an intelligible fashion and written in standard English?

Reviewer #1: Yes

Reviewer #2: (No Response)

6. Review Comments to the Author

Reviewer #1: My suggestions gave in the previous statge are all solved in this version. Thus, I would like to suggest the MS can be welcomed to the journal.

Reviewer #2: The authors have addressed all my concerns very well, I'd like to recommend it for publication for its current form

7. PLOS authors have the option to publish the peer review history of their article (what does this mean?). If published, this will include your full peer review and any attached files.

Reviewer #1: No

Reviewer #2: No

---

## [Editor Report · Acceptance letter]

24 Jul 2023

PONE-D-23-12612R1 

Network topology and movement cost, not updating mechanism, determine the evolution of cooperation in mobile structured populations 

Dear Dr. Broom:

I'm pleased to inform you that your manuscript has been deemed suitable for publication in PLOS ONE. Congratulations! Your manuscript is now with our production department. 

Kind regards, 

on behalf of

Prof. Jun Tanimoto 

Academic Editor

PLOS ONE